# Polynomial Semantics of Tractable Probabilistic Circuits

**Oliver Broadrick**[1]         **Honghua Zhang**[1]         **Guy Van den Broeck**[1]

[1]Computer Science Dept., University of California, Los Angeles, California, USA

## Abstract

Probabilistic circuits compute multilinear polynomials that represent multivariate probability distributions. They are tractable models that support efficient marginal inference. However, various polynomial semantics have been considered in the literature (e.g., network polynomials, likelihood polynomials, generating functions, and Fourier transforms). The relationships between circuit representations of these polynomial encodings of distributions is largely unknown. In this paper, we prove that for distributions over binary variables, each of these probabilistic circuit models is equivalent in the sense that any circuit for one of them can be transformed into a circuit for any of the others with only a polynomial increase in size. They are therefore all tractable for marginal inference on the same class of distributions. Finally, we explore the natural extension of one such polynomial semantics, called probabilistic generating circuits, to categorical random variables, and establish that inference becomes #P-hard.

## 1 INTRODUCTION

Modeling probability distributions in a way that allows efficient probabilistic inference (e.g. computing marginal probabilities) is a key challenge in machine learning. Much research towards meeting this challenge has led to the development of families of tractable models including bounded-treewidth graphical models such as hidden Markov models [Rabiner and Juang, 1986], (mixtures of) Chow-Liu Trees [Chow and Liu, 1968, Meila and Jordan, 2000, Selvam et al., 2023], determinantal point processes (DPPs) [Kulesza and Taskar, 2012], and various families of probabilistic circuits (PCs) such as sum-product networks [Poon and Domingos, 2011, Peharz et al., 2018] and probabilistic sen-

tential decision diagrams [Kisa et al., 2014]. As a unifying representation for all aforementioned models, probabilistic circuits (PCs) compactly represent multilinear polynomials that encode probability distributions (Fig. 2) [Choi et al., 2020]. However, multiple semantics of the polynomials represented by a PC have been considered, and their relationships are largely unknown. We study these various semantics and show that for binary variables they are all equivalent in the sense that a circuit in one semantics can be tranformed into a circuit in any of the others with at most a polynomial change in size.[1]

The simplest polynomial encoding of a probability distribution, which we call the *likelihood polynomial*, directly computes the probability mass function [Roth and Samdani, 2009]. The standard polynomial used in the PC literature, called the *network polynomial* [Darwiche, 2003], uses additional input variables and special structure to enable computation of arbitrary marginal probabilities. While it is not obvious whether the circuit representation of a likelihood polynomial should support marginal inference, we provide a linear-time inference algorithm. We also give a polynomial-time transformation from likelihood polynomial circuits to network polynomial circuits, using a classic circuit complexity result of Strassen [1973] to enable divisions.

Probability generating functions (*generating polynomials* for short) have also been considered as semantics for circuits and support efficient marginal inference [Zhang et al., 2021, Harviainen et al., 2023]. While it is straightforward to transform a circuit computing a network polynomial to a circuit computing a generating polynomial, the other direction is unclear. For example, there are distributions that can be succinctly expressed by circuits computing generating

---

[1]Some of the results in this paper were found independently by Agarwal and Bläser [2024] and submitted to a conference around the same time; they provide the circuit transformation from generating polynomials to network polynomials (our Theorem 3) and show that inference in categorical generating circuits is #P-hard (our Theorem 6).

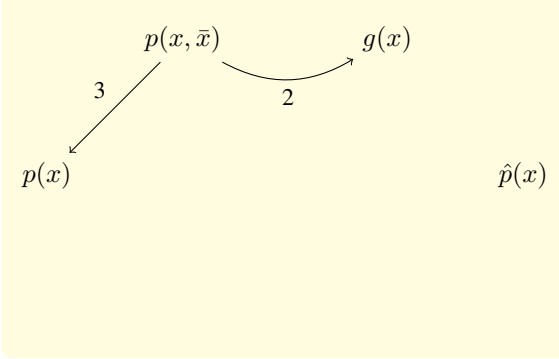
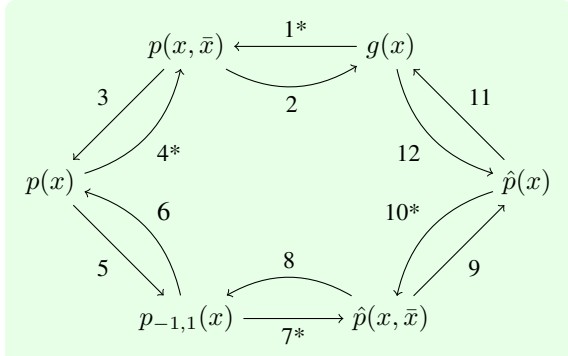

Figure 1: Polynomial time circuit transformations between polynomial semantics including: likelihood $p(x)$, network $p(x, \bar{x})$, generating $g(x)$, and Fourier $\hat{p}(x)$ polynomials. Previously known transformations are displayed on the left; (2) is given in [Zhang et al., 2021], and (3) is implicit in [Roth and Samdani, 2009]. The results in this paper are shown on the right. Edges labeled by * correspond to transformations which map circuits of size $s$ to circuits of size $O(sn^2)$; other edges correspond to transformations which map circuits of size $s$ to circuits of size $O(s)$.

polynomials but not by decomposable[2] circuits computing network polynomials that use only positive weights [Zhang et al., 2021, 2020, Martens and Medabalimi, 2015]. However, we present a polynomial-time transformation from generating polynomial semantics to network polynomial semantics again using the result of Strassen [1973].

Lastly, the Fourier transform of the probability mass function (called characteristic function in probability theory) has been considered for inference in graphical models [Xue et al., 2016] and as a semantics for circuits [Yu et al., 2023]. We find simple linear-time transformations between circuits computing this Fourier transform and circuits computing the generating polynomial. This connects all aforementioned semantics (likelihood, network, generating, and Fourier polynomials) by polynomial-time transformations; Figure 1 summarizes the transformations. Consequently, any distribution which can be succinctly expressed in one semantics can be succinctly expressed in them all, including for instance DPPs [Kulesza and Taskar, 2012] which were previously only known to be succinctly expressible using generating polynomials [Zhang et al., 2021].

Our transformations make no assumptions on the structure of the circuit. However, it is common to use syntactic constraints on a circuit to guarantee desired semantic properties (e.g. a circuit with nonnegative weights and constants computes a polynomial with nonnegative coefficients), while possibly sacrificing succinctness [de Colnet and Mengel, 2021]. In particular, PCs are typically assumed to be decomposable – children of product nodes contain disjoint sets of variables [Darwiche and Marquis, 2002] – to guarantee that the computed polynomial is multilinear. While our transformations hold regardless of syntactic properties, we show in Section 6 that they simplify for decomposable circuits.

---

[2]A standard structural property discussed in Section 6.

Finally, we extend our discussion to categorical distributions. While there is a specialized inference algorithm for circuits computing generating polynomials for binary distributions, generating polynomials are well-defined for arbitrary categorical distributions. Accordingly, in Section 7 we consider the problem of inference on a circuit computing a generating polynomial with $k$ categories and show that for $k \geq 4$ inference is #P-hard.

## 2 BACKGROUND

We use the mass function $\mathrm{Pr} : \{0, 1\}^n \to \mathbb{R}$ to specify a probability distribution on $n$ binary random variables $\boldsymbol{X} = \{X_1, X_2, \ldots, X_n\}$, each taking values in $\{0, 1\}$. We denote $[n] = \{1, 2, \ldots, n\}$. We use $\boldsymbol{x}$ to denote an assignment to the random variables, and for any $S \subseteq [n]$, we let $\boldsymbol{x}_S$ denote the assignment $X_i = 1$ for $i \in S$ and $X_i = 0$ for $i \notin S$. We study polynomials in indeterminates $x_1, \ldots, x_n$ which we abbreviate to $x$. A polynomial is *multilinear* if it is linear in every variable. For example, the polynomials $x_1 x_3 - x_2 x_3$ and $x_1 x_2 x_3 + 1$ are multilinear, but $x^2$ and $x_1 x_2^7 + 1$ are not.

In this paper we consider multilinear polynomials as representations of probability distributions. To compactly represent polynomials, we use arithmetic circuits, a fundamental object of study in computer science [Shpilka and Yehudayoff, 2010] which have proven useful for representing tractable probabilistic models.

**Definition 1.** *An arithmetic circuit (AC) is a directed acyclic graph consisting of three types of nodes:*

1. *Sum nodes $\oplus$ with weighted edges to children;*

2. *Product nodes $\otimes$ with unweighted edges to children;*

3. *Leaf nodes, which are variables in $\{x_1, \ldots, x_n\}$ or constants in $\mathbb{R}$.*

*An AC has one node of in-degree zero, and we refer to it as the* root. *The size of an AC is the number of edges in it.*

Each node in an AC represents a polynomial: (i) each leaf represents the polynomial $x_i$ or a constant, (ii) each sum node represents the weighted sum of the polynomials represented by its children, and (iii) each product node represents the product of the polynomials represented by its children. The polynomial represented by an AC is the polynomial represented by its root. We note that the standard definition of AC in the circuit complexity literature uses unweighted sums, but the models are equivalent up to constant factors. For the remainder of this paper we use the term circuit to mean arithmetic circuit.

Note that when we say that two polynomials/circuits are the same, we *do not* mean that they agree on all inputs in $\{0,1\}^n$ but that they agree on all real inputs in $\mathbb{R}^n$; the polynomials are identical elements in the ring of polynomials $\mathbb{R}[x_1, \ldots, x_n]$. Moreover, we note that while a polynomial may be large (e.g. containing exponentially many nonzero monomials) there may be a much more succinct circuit computing the polynomial (e.g. of polynomial size in the number of variables).

# 3 NETWORK AND LIKELIHOOD POLYNOMIALS

There are various polynomials containing all the information of a binary distribution $\Pr$, in the sense that any value $\Pr(\boldsymbol{x})$ can be recovered from the polynomial alone. It is known that efficient circuit representations of some such polynomials allow tractable marginal inference, but a unified analysis of the various polynomial representations is lacking. In this section, we begin with the most studied such polynomial, the network polynomial, and establish its connections to the more natural – yet still, as we show, tractable – likelihood polynomial.

## 3.1 NETWORK POLYNOMIALS

Darwiche [2003] showed that Bayesian Networks can be compiled to circuits computing a certain polynomial representation of their distribution which he called the *network polynomial* (also see Castillo et al. [1995]). The *network polynomial* of binary probability distribution $\Pr$ is

$$p(x_1, \ldots, x_n, \bar{x}_1, \ldots, \bar{x}_n) = \sum_{S \subseteq [n]} \Pr(\boldsymbol{x}_S) \prod_{i \in S} x_i \prod_{i \notin S} \bar{x}_i.$$

Significant work towards learning and applying circuits computing this polynomial has since been developed [Poon and Domingos, 2011, Peharz et al., 2020, Liu et al., 2021]. In particular, this is the canonical polynomial computed by circuits in the growing literature on Probabilistic Circuits

(PC) [Choi et al., 2020]. A key feature of circuits computing network polynomials is that they enable simple linear-time marginal inference. Note that while the algorithm for marginalization is typically given for smooth and decomposable circuits, the following proposition holds for circuits of any structure which compute a network polynomial.

**Proposition 1.** *Computing marginals on a circuit of size $s$ representing a network polynomial takes $O(s)$ time. For the random variable assignment $X_i = 1$, set $x_i = 1$ and $\bar{x}_i = 0$; for $X_i = 0$, set $x_i = 0$ and $\bar{x}_i = 1$; marginalize out $X_i$ by setting $x_i = \bar{x}_i = 1$. Evaluating the circuit now computes the marginal probability.*

The network polynomial has a very specific structure. First, it is multilinear. Second, every nonzero monomial contains either $x_i$ or $\bar{x}_i$ for *every* $i \in \{1, 2, \ldots, n\}$. We wonder whether this structure of the monomials with variables $x_i$ and $\bar{x}_i$ is necessary for marginal inference. We next consider a simple polynomial which does not use this structure with the $\bar{x}_i$ variables, but as we show, still remains tractable.

## 3.2 LIKELIHOOD POLYNOMIALS

Roth and Samdani [2009] considered perhaps the simplest polynomial representation of $\Pr$, that which directly computes $\Pr$ using variables $x_1, \ldots, x_n$. Such a polynomial can be obtained from a network polynomial by substituting $\bar{x}_i = 1 - x_i$ (transformation 3 in Figure 1). We call this the *likelihood polynomial*:

$$p(x_1, \ldots, x_n) = \sum_{S \subseteq [n]} \Pr(\boldsymbol{x}_S) \prod_{i \in S} x_i \prod_{i \notin S} (1 - x_i). \quad (1)$$

While conceptually simple, it is not clear how or whether it is possible to efficiently compute marginals given a circuit representation of the likelihood polynomial. In particular, Roth and Samdani [2009] considered only "flat" representations of the likelihood polynomial, where all monomials with nonzero coefficients are stored explicitly. While marginal inference is linear in the size of the flat representation, there is an exponential gap in the succinctness of circuits and flat representations.

We note that both network polynomials and likelihood polynomials are multilinear. Indeed, inference on circuits that agree with $\Pr$ on all inputs in $\{0,1\}^n$ (like likelihood polynomials) is intractable without multilinearity. For example, even if we just allow circuits computing polynomials that are quadratic in each variable, marginal inference is already #P-hard (e.g., implicit in the proof of Theorem 2 in Khosravi et al. [2019]).

We show that on a circuit representing a likelihood polynomial, marginal probabilities can be computed in linear time.

**Proposition 2.** *Marginal probabilities on a circuit of size $s$ representing a likelihood polynomial can be computed in time $O(s)$.*

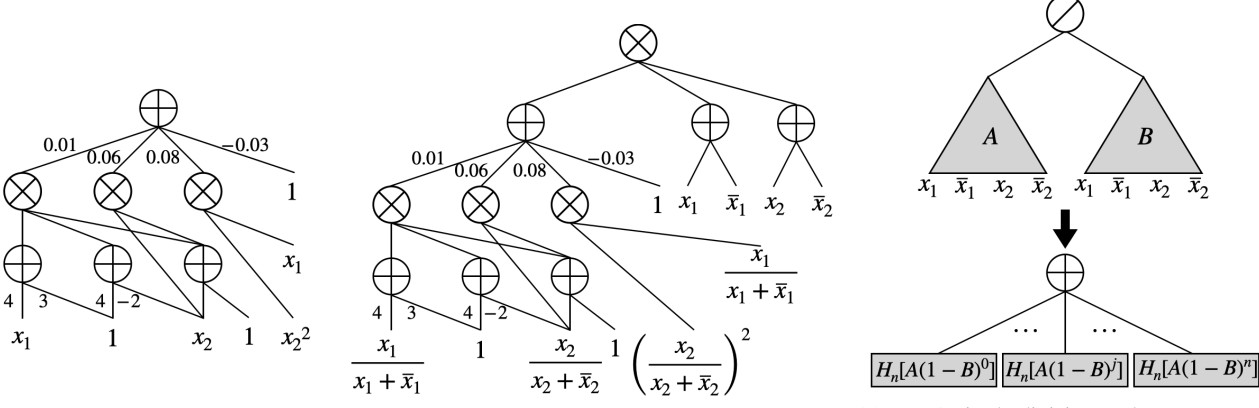

(a) PC for likelihood polynomial.

(b) Leaf nodes replaced with division gadgets.

(c) top: A single division node.
bottom: Sum of homogeneous parts.

Figure 2: An example transforming a circuit representing a likelihood polynomial $p(x) = 0.08x_1x_2 + 0.16x_1 + 0.12x_2 + 0.09$ to a circuit representing a network polynomial. First, (b) gadgets using division nodes are introduced at the leaves (as well as a multiplying factor) to obtain a rational function equivalent to the network polynomial. Then (c:top) all divisions are pushed to a single division node at the root so $p(x, \bar{x}) = A/B$, and (c:bottom) a sum over necessary homogeneous parts of $A$ and $B$ is formed.

*Proof.* Consider the following expression where $p$ denotes the likelihood polynomial (Eq. 1).

$$\left( \prod_{i=1}^{n} (x_i + \bar{x}_i) \right) p \left( \frac{x_1}{x_1 + \bar{x}_1}, \dots, \frac{x_n}{x_n + \bar{x}_n} \right) \qquad (2)$$

This expression simplifies as follows:

$$\left( \prod_{i=1}^{n} (x_i + \bar{x}_i) \right) \cdot$$

$$\sum_{S \subseteq [n]} \Pr(\boldsymbol{x}_s) \prod_{i \in S} \frac{x_i}{x_i + \bar{x}_i} \prod_{i \notin S} \left( 1 - \frac{x_i}{x_i + \bar{x}_i} \right)$$

$$= \sum_{S \subseteq [n]} \Pr(\boldsymbol{x}_s) \prod_{i \in S} x_i \prod_{i \notin S} \bar{x}_i$$

$$= p(x, \bar{x}).$$

Therefore, this expression[3] (2) behaves functionally as the network polynomial, and so any marginal probability can be computed as in Proposition 1. $\square$

Moreover, this expression (Eq. 2) naturally corresponds to a circuit computing the network polynomial *using division nodes*; starting with a circuit computing the likelihood polynomial, replace inputs $x_i$ with $x_i/(x_i + \bar{x}_i)$ and multiply the whole circuit by $\prod_{i=1}^{n}(x_i + \bar{x}_i)$. However, the PC literature does not typically use division nodes, and available software libraries and known algorithms would need to be reconsidered to use division nodes, not to mention possible

---

[3]Readers familiar with the weighted model counting task on decomposable logic circuits might recognize a neutral labeling function in this expression [Kimmig et al., 2017].

divide-by-zero problems – which will arise in Section 4. This leads to the question, can we find a circuit computing an equivalent polynomial without use of division nodes? Classic work in the circuit complexity literature by Strassen [1973] provides a positive answer.

**Theorem 1** (Strassen). *If there is an arithmetic circuit of size $s$ with division nodes computing polynomial $p$ of degree $d$ in $n$ variables over an infinite field, then there exists an arithmetic circuit of size $poly(s, d, n)$ that computes $p$ using only addition and multiplication nodes.*

Therefore we have the following theorem, corresponding to transformation 4 in Figure 1.

**Theorem 2.** *Let $\Pr$ be a probability distribution on $n$ binary random variables. Then a circuit of size $s$ computing the likelihood polynomial for $\Pr$ can be transformed to a circuit of size $O(sn^2)$ computing the network polynomial for $\Pr$.*

To illustrate the algorithm, we consider the example in Figure 2. Figure 2a shows the initial circuit that represents the likelihood polynomial. Figure 2b shows the circuit computing the expression with division nodes. To remove divisions, the first observation is that all division nodes can be moved 'up' to a single division at the output node using the identities $(a/b) \times (c/d) = (ac)/(bd)$ and $(a/b) + (c/d) = (ad + bc)/(bd)$, as visualized in Figure 2c. At this point we have the network polynomial written as a ratio of two polynomials, $p(x, \bar{x}) = A(x, \bar{x})/B(x, \bar{x})$. Without loss of generality we assume $B$ has constant term one, i.e. $B(0, 0, \dots, 0) = 1$. If $B$ does not already have constant term 1, then its inputs can be translated and the whole function scaled as needed.

One additional result from the circuit complexity literature is needed at this point; for any circuit $f$ of size $s$ and degree $d$, a circuit of size $O(d^2 s)$ can be constructed (with $d + 1$ outputs) computing $H_0[f], H_1[f], \ldots, H_d[f]$ where $f = \sum_i H_i[f]$, and each $H_i[f]$ is *homogeneous* meaning that every monomial of $H_i[f]$ has degree $i$ [Shpilka and Yehudayoff, 2010]. This process is called *homogenization*, and the $H_i[f]$'s the *homogeneous parts* of $f$.

The final division node can now be eliminated by use of the common identity $\frac{a}{1-r} = \sum_{j=0}^{\infty} ar^j$. We have

$$p(x, \bar{x}) = \frac{A}{B} = \frac{A}{1 - (1 - B)} = \sum_{j=0}^{\infty} A(1 - B)^j.$$

In particular, these equalities hold for the homogeneous parts of $p(x, \bar{x})$. And, because $B(0, \ldots, 0) = 1$, we know that $1 - B$ has constant term zero, and so all monomials in $(1 - B)^j$ have degree at least $j$. Since we know that the network polynomial $p(x, \bar{x})$ has all terms of degree exactly $n$, we only need to compute

$$H_n[p(x, \bar{x})] = \sum_{j=0}^{n} H_n[A(1 - B)^j],$$

as illustrated by Figure 2c. In particular, a single circuit computing $(1 - B)^j$ for $j \in \{0, 1 \ldots, n\}$ can be homogenized in addition to homogenizing $A$, to compute $p(x, \bar{x})$ with size $O(sn^2)$.

We note that the circuit obtained by this transformation contains negative parameters.

## 4 GENERATING POLYNOMIALS

So far we have considered circuits that directly compute a distribution. However, there are other well known polynomial representations of probability distributions which have been shown as promising representations for tractable probabilistic modeling. Zhang et al. [2021] consider circuits computing the *probability generating function* of a distribution. Generating functions are well studied in mathematics as theoretical objects [Wilf, 2005], but have recently been identified as useful data structures [Zhang et al., 2021, Klinkenberg et al., 2023, Zaiser et al., 2023]. The *generating polynomial* for probability distribution $\Pr$ is

$$g(x_1, \ldots, x_n) = \sum_{S \subseteq [n]} \Pr(\boldsymbol{x}_s) \prod_{i \in S} x_i.$$

Zhang et al. [2021] call circuits computing generating polynomials Probabilistic Generating Circuits (PGCs) and show that marginal inference on PGCs is tractable. For a PGC of size $s$ in $n$ variables, they provide an $O(sn \log n \log \log n)$ marginal inference algorithm which has been improved by Harviainen et al. [2023] to $O(sn)$. It is also noted by Zhang

et al. [2021] that circuits computing network polynomials can be transformed to PGCs simply by replacing $\bar{x}_i$'s by 1, and so any distribution with a polynomial-size circuit computing its network polynomial also has a polynomial-size PGC; this is transformation 2 in Figure 1. On the other hand, they show that there are distributions with polynomial-size PGCs but for which any decomposable circuit computing the network polynomial using only positive weights has exponential size (and additional PGC lower bounds are known [Bläser, 2023]). It is left as an open question whether this separation still holds for circuits with unrestricted weights. We provide a negative answer. Using a method similar to that in Section 3, we show that given a PGC, one can find a circuit computing the network polynomial with a polynomial increase in size; this is transformation 1 in Figure 1.

**Theorem 3.** *Let* $\Pr$ *be a probability distribution on* $n$ *binary random variables. Then a circuit of size* $s$ *computing the probability generating function for* $\Pr$ *can be transformed to a circuit of size* $O(sn^2)$ *computing the network polynomial for* $\Pr$.

*Proof.* We obtain the desired circuit by first constructing a circuit that computes $p(x, \bar{x})$ using division nodes. Observe

$$\left( \prod_{i=1}^{n} \bar{x}_i \right) g \left( \frac{x_1}{\bar{x}_1}, \frac{x_2}{\bar{x}_2}, \ldots, \frac{x_n}{\bar{x}_n} \right)$$

$$= \left( \prod_{i=1}^{n} \bar{x}_i \right) \sum_{S \subseteq [n]} \Pr(\boldsymbol{x}_S) \prod_{i \in S} \frac{x_i}{\bar{x}_i}$$

$$= \sum_{S \subseteq [n]} \Pr(\boldsymbol{x}_S) \prod_{i \in S} x_i \prod_{i \notin S} \bar{x}_i$$

$$= p(x_1, \ldots, x_n, \bar{x}_1, \ldots, \bar{x}_n)$$

The degree of $p(x, \bar{x})$ is $n$, and so using Theorem 1 as in Section 3, there is a circuit computing $p(x, \bar{x})$ without division nodes of size $O(sn^2)$. $\square$

We note the similarity of the proof of Theorem 2 to that of Theorem 3. They both involve constructing a circuit to represent $p(x, \bar{x})$ initially using division nodes and then removing the division nodes. We also note the crucial difference between the proofs; in the construction for Theorem 3, the circuit with division nodes *can not be used to evaluate* $p(x, \bar{x})$ *directly because it would require division by zero whenever* $\bar{x}_i = 0$ *for any* $i \in [n]$. Therefore the ability to remove divisions while maintaining equivalence of the polynomial computed is essential for this transformation to be meaningful. As one immediate consequence, this implies the existence of polynomial size PCs computing network polynomials for DPPs since Zhang et al. [2021] showed the existence of polynomial size PGCs for DPPs. Another practical benefit is that rather than using a bespoke polynomial-interpolation algorithm for inference in PGCs [Harviainen et al., 2023], there is a simple feedforward (and

easily implemented, on a GPU for example) method of inference for PGCs after the transformation has been performed.

# 5 FOURIER TRANSFORMS

Fourier analysis involves representing functions in the frequency domain and is ubiquitous across math and computer science. Yu et al. [2023] show that circuits representing Fourier transforms (called *characteristic functions* in probability theory) can improve learning in a mixed discrete-continuous setting while still supporting marginal inference when the circuit is smooth and decomposable (see Section 6 for discussion of these properties). Xue et al. [2016] use Fourier representations for inference in graphical models too. The *Fourier transform* [O'Donnell, 2014] of pseudoboolean function $p : \{0,1\}^n \to \mathbb{R}$ is the function $\hat{p} : \{0,1\}^n \to \mathbb{R}$ given by

$$\hat{p}(x) = 2^{-n} \sum_{v \in \{0,1\}^n} p(v)(-1)^{\langle v,x \rangle}$$

where $\langle v, x \rangle$ is the standard inner (dot) product over the reals. It is convenient that in this binary case, $\hat{p}(x)$ can also be simply written as a multilinear polynomial (note that the equality holds on its domain $\{0,1\}^n$):

$$\hat{p}(x) = 2^{-n} \sum_{S \subseteq [n]} p(v_S) \prod_{i \in S} (1 - 2x_i) \qquad (3)$$

where $v_S$ is the element of $\{0,1\}^n$ with $v_i = 1$ for $i \in S$ and $v_i = 0$ for $i \notin S$. To see this, identify $S$ with $v_S$, and then $(-1)^{\langle v_S, x \rangle} = (-1)^{\sum_{i \in S} x_i} = \prod_{i \in S}(-1)^{x_i} = \prod_{i \in S}(1 - 2x_i)$. For the rest of the paper we use $\hat{p}(x)$ to refer to this multilinear polynomial (Eq. 3). We note that Fourier analysis of binary functions is a rich subject in its own right and refer the reader to [O'Donnell, 2014].

While there is no obvious connection between network polynomials, generating functions, and Fourier transforms, we show that they are in fact closely related. This relation hinges on switching between the domains $\{0,1\}^n$ and $\{-1,1\}^n$. Accordingly, we define for any multilinear polynomial $f$ its counterpart $f_{-1,1}$ as follows:

$$f_{-1,1}(x_1, \ldots, x_n) = f\left(\frac{1-x_1}{2}, \ldots, \frac{1-x_n}{2}\right), \quad (4)$$

also a multilinear polynomial. Similarly, observe that

$$f(x_1, \ldots, x_n) = f_{-1,1}(1 - 2x_1, \ldots, 1 - 2x_n). \quad (5)$$

Note that $f$ and $f_{-1,1}$ compute *the same function* on the respective domains $\{0,1\}^n$ and $\{-1,1\}^n$ up to the bijection $\phi : \{0,1\} \to \{-1,1\}$ given by $\phi(b) = (-1)^b$ applied bitwise. In particular, Equations 4 and 5 can be applied to circuits with modifications at only the leaves, giving the following lemma.

**Lemma 1.** *A circuit of size $s$ computing polynomial $f$ (respectively $f_{-1,1}$) can be transformed to a circuit of size $O(s)$ computing $f_{-1,1}$ (respectively $f$).*

We now make a simple observation that connects Fourier transforms with generating polynomials; up to a constant factor, generating polynomials *are* Fourier transforms, written on the domain $\{-1,1\}^n$.

**Proposition 3.** *Let $\Pr$ be a probability distribution on $n$ binary random variables with generating polynomial $g(x)$ and Fourier polynomial $\hat{p}_{-1,1}(x)$ on the domain $\{-1,1\}$. Then $g(x) = 2^n \hat{p}_{-1,1}(x)$.*

*Proof.*

$$g(x) = \sum_{S \subseteq [n]} \Pr(\boldsymbol{x}_S) \prod_{i \in S} x_i$$

$$= \sum_{S \subseteq [n]} \Pr(\boldsymbol{x}_S) \prod_{i \in S} \left(1 - 2\left(\frac{1-x_i}{2}\right)\right)$$

$$= 2^n \hat{p}_{-1,1}(x). \qquad \square$$

Using only the ability to switch between the domains $\{0,1\}^n$ and $\{-1,1\}^n$ and Proposition 3, we now have transformations 11 and 12 in Figure 1.

**Theorem 4.** *Let $\Pr$ be a probability distribution on $n$ binary random variables. Then a circuit of size $s$ computing the generating polynomial $g(x)$ (respectively $\hat{p}(x)$) for $\Pr$ can be transformed to a circuit of size $O(s)$ representing the Fourier transform $\hat{p}(x)$ (respectively $g(x)$) for $\Pr$.*

*Proof.* Proposition 3 and Lemma 1. $\square$

Having observed this connection between generating polynomials and Fourier polynomials, we have completed a path between $p(x)$ and $\hat{p}(x)$ in Figure 1 (the upper half), i.e. a polynomial-time transformation between circuits computing them. However, we observe that this path more naturally corresponds to computing the inverse Fourier transform, and there is a symmetric set of transformations that compute $\hat{p}(x)$ from $p(x)$ in a more natural way. These transformations will form the lower half of Figure 1. We now give the more standard definition of the Boolean Fourier transform $\hat{p}(x)$, in which the values of $\hat{p}(x)$ are the coefficients needed to write $p(x)$ as a linear combination of parity functions [O'Donnell, 2014, Thm 1.1]:

$$p(x) = \sum_{S \subseteq [n]} \hat{p}(v_S)(-1)^{\sum_{i \in S} x_i}$$

$$= \sum_{S \subseteq [n]} \hat{p}(v_S) \prod_{i \in S}(1 - 2x_i)$$

where equalities hold on the domain $\{0,1\}^n$. The values $\{\hat{p}(v_S)\}_{S \subseteq [n]}$ are called the *Fourier coefficients*, or, collectively, the *Fourier spectrum* of $p(x)$. Moreover, this is

more naturally written on the domain $\{-1, 1\}^n$ due to the equivalence of the parity functions $\prod_{i \in S}(1 - 2x_i)$ to the monomials $\prod_{i \in S} x_i$ on the respective domains $\{0, 1\}^n$ and $\{-1, 1\}^n$. That is, we have

$$p_{-1,1}(x) = \sum_{S \subseteq [n]} \hat{p}(v_S) \prod_{i \in S} x_i. \qquad (6)$$

Now we recall the transformation from $g(x)$ to $p(x, \bar{x})$ (Theorem 3 and transformation 1 in Figure 1). In general, this transformation takes a polynomial in indeterminates $x_1, \ldots, x_n$ with monomials $\left\{c_S \prod_{i \in S} x_i\right\}_{S \subseteq [n]}$ and produces a polynomial in indeterminates $x_1, \ldots, x_n, \bar{x}_1, \ldots, \bar{x}_n$ with monomials $\left\{c_S \prod_{i \in S} x_i \prod_{i \notin S} \bar{x}_i\right\}_{S \subseteq [n]}$; this latter polynomial allows us to 'extract' the coefficients of the former by the substitution $\bar{x}_i = 1 - x_i$. As seen in Eq 6, we now have an identical problem where we would like to compute $\hat{p}(x)$ from $p_{-1,1}(x)$, i.e., to 'extract' the Fourier coefficients from $p_{-1,1}(x)$. So, we analogously define the polynomial $\hat{p}(x, \bar{x})$

$$\hat{p}(x, \bar{x}) = \sum_{S \subseteq [n]} \hat{p}(v_S) \prod_{i \in S} x_i \prod_{i \notin S} \bar{x}_i$$

and obtain transformations 7 and 8 in Figure 1 (by Theorem 3). Moreover, by definition, the relationship between $\hat{p}(x, \bar{x})$ and $\hat{p}(x)$ is identical to that between $p(x, \bar{x})$ and $p(x)$, and so transformations 9 and 10 in Figure 1 follow (i.e., using Theorem 2).

Having now completed the transformations presented in Figure 1, we ask how they simplify in the presence of structural constraints common in the PC literature.

# 6 DECOMPOSABILITY

So far we make no assumptions on the structural properties of circuits; in this section, we consider the special case where the circuit is *decomposable* [Darwiche and Marquis, 2002], which is a common assumption that guarantees tractable marginal inference. We show that in this case some of the transformations described before can be simplified. We use the *scope* of a node to refer to the set of all $i$ such that variables $x_i$ or $\bar{x}_i$ appear as inputs among its descendants and itself.

**Definition 2** (Decomposability). *A product node is decomposable if its children have disjoint scopes. A circuit is decomposable if all its product nodes are decomposable.*[4]

**Definition 3** (Smoothness). *A sum node in indeterminates $x$ and $\bar{x}$ is smooth if its children have the same scope. A circuit is smooth if all of its sum nodes are smooth.*

---

[4]This property is called syntactic multilinearity in the arithmetic circuits literature.

Decomposability is a very common property because it guarantees multilinearity and, when paired with smoothness, ensures tractable marginal inference. In particular, it is well known that if a circuit is smooth and decomposable, it computes a network polynomial [Poon and Domingos, 2011, Peharz et al., 2015, Choi et al., 2020]. We note that if a circuit is decomposable, then it can be made smooth efficiently (increasing the size at most by a linear factor [Choi et al., 2020] and less for certain decomposable structures Shih et al. [2019]).

We now show how the transformations used for Theorems 2, 3, and 4 can be simplified for decomposable circuits. First, we show that in decomposable circuits Fourier transforms correspond to trivial modifications *at only the leaves*.

**Theorem 5.** *A decomposable circuit of size $s$ representing a likelihood polynomial $p(x)$ can be transformed to a decomposable circuit of size $O(s)$ representing its Fourier transform $\hat{p}(x)$ by only modifying the leaves.*

We give a full proof of Theorem 5 in the Appendix.

*Proof Sketch.* A circuit representing $\hat{p}(x)$ can be constructed with modifications pushed to the leaves inductively. Decomposability enables pushing past product nodes; linearity of the Fourier transform enables pushing past sum nodes. Leaf nodes are univariate and so can be transformed directly. $\square$

Transformations 1 and 4 in Figure 1 can be simplified when the initial circuits are decomposable; the decomposability is preserved during the transformation, and the worst-case increase in size is lowered to $O(sn)$. First, a decomposable circuit of size $s$ computing a likelihood polynomial $p(x)$ can be transformed to decomposable circuit of size $O(sn)$ computing $p(x, \bar{x})$. We note that this problem is exactly that of smoothing [Darwiche, 2000, Shih et al., 2019] and so the following lemma is included for completeness but is already known. In particular, this shows how Theorem 2 can be viewed as a generalization of smoothing to circuits that are not decomposable.

**Lemma 2** (Darwiche [2000], Shih et al. [2019]). *A decomposable circuit of size $s$ computing likelihood polynomial $p(x)$ can be transformed to a decomposable circuit of size $O(sn)$ computing network polynomial $p(x, \bar{x})$.*

Also, a decomposable circuit of size $s$ computing a generating polynomial $g(x)$ can be transformed to decomposable circuit of size $O(sn)$ computing $p(x, \bar{x})$. This problem is very similar to smoothing and can be solved in the same way; rather than smoothing with gadgets computing $x_i + \bar{x}_i$, simply use $\bar{x}_i$.

**Lemma 3.** *A decomposable circuit of size $s$ computing generating polynomial $g(x)$ can be transformed to a decompos-*

$$\begin{pmatrix} 0 & 1 & 0 & 0 \\ 0 & 1 & 0 & 0 \\ 0 & 1 & 0 & 0 \\ 0 & 1 & 0 & 0 \end{pmatrix} \rightarrow \begin{pmatrix} 0 & 1 & 0 & 0 & 0 \\ 0 & 0 & 0 & 0 & \underline{1} \\ 0 & 0 & 0 & 0 & \underline{1} \\ 0 & 1 & 0 & 0 & 0 \\ 0 & \underline{1} & 0 & 0 & \underline{1} \end{pmatrix}$$

Figure 3: An example of the permanent-preserving operation used to make $M$ sparse. The new row and column are shaded in blue. The newly-added nonzero entries that preserve the permanent of the matrix are singly-underlined. The two nonzero entries that moved from their original column (highlighted in orange) to the new one are doubly-underlined. The number of nonzero entries in the second column has decreased by one.

*able circuit of size $O(sn)$ computing network polynomial $p(x, \bar{x})$.*

We note that Lemmas 2 and 3 hold also for the symmetric transformations as described in Section 5, i.e., for decomposable versions of transformations 7 and 10 in Figure 1.

# 7 CATEGORICAL DISTRIBUTIONS

So far we have considered binary probability distributions with probability mass functions of the form $\Pr : \{0, 1\}^n \to \mathbb{R}$. Of course, categorical distributions with mass functions of the form $\Pr : S^n \to \mathbb{R}$ for an arbitrary finite set $S$ are also of interest. In the PC literature, categorical distributions are typically encoded as binary distributions using binary indicator variables [Darwiche, 2003, Poon and Domingos, 2011, Choi et al., 2020]. Indeed, network polynomials have no other obvious extension to the categorical setting; however, generating polynomials do. In fact, the generating polynomials considered in [Zhang et al., 2021] are a restriction to the binary case of the following more general and standard definition. Let $\Pr : K^n \to \mathbb{R}$ be a probability mass function with $K = \{0, 1, 2, \ldots, k - 1\}$. Then the probability generating polynomial of $\Pr$ is

$$g(x) = \sum_{(d_1, d_2, \ldots, d_n) \in K^n} \Pr(d_1, \ldots, d_n) x_1^{d_1} x_2^{d_2} \cdots x_n^{d_n}.$$

It is natural then to consider a categorical Probabilistic Generating Circuit (PGC) as a circuit computing the generating polynomial of a categorical distribution with more than two categories. This begs the question, are categorical PGCs a tractable model in general? To this, we give a negative answer. In fact for variables with $k \geq 4$ categories, not only is marginal inference hard, but even computing likelihoods is hard.

**Theorem 6.** *Computing likelihoods on a categorical PGC is #P-hard for $k \geq 4$ categories.*

Here we provide a sketch of our proof of Theorem 6, giving the full proof in the Appendix. Our proof is a reduction from $\{0, 1\}$-permanent to categorical PGC inference. The classic work of Valiant [1979b] shows that computing the permanent of matrices with entries in $\{0, 1\}$ is #P-complete. The permanent of a matrix $M$ is

$$\text{per } M = \sum_{\sigma \in S_n} \prod_{i=1}^{n} M_{i, \sigma(i)}$$

where $S_n$ is the symmetric group of order $n$. Our reduction proceeds in two steps.

First, we find a (modestly) larger matrix $M'$ with the same permanent as $M$ and with the sparsity property that every column contains at most three nonzero entries. To do so, let $M \in \{0, 1\}^{n \times n}$, and assume the $t$th column of $M$ still has more than 3 nonzero entries. Add a new $(n + 1)$th row and column to the matrix with main diagonal entry 1, and with the $t$th entry of the new row 1, and all other entries zero. We are now free to 'move' any two nonzero entries from the $t$th column to the new $(n + 1)$th column without changing the permanent of the matrix. Figure 3 shows an example of this sparsifying, permanent-preserving operation. The number of nonzero entries in the $t$th column has now decreased by one (and the new column has only three nonzero entries), and so repeating this operation at most $n^2$ times will yield the desired matrix $M'$.

Second, we use a polynomial construction from Valiant [1979a] (and Koiran and Perifel [2007]):

$$g(x_1, \ldots, x_n) = \prod_{i=1}^{n'} \sum_{j=1}^{n'} M'_{i,j} x_j.$$

The coefficient of the monomial $\prod_{i=1}^{n'} x_i$ in $g(x)$ is exactly $\text{per } M' = \text{per } M$. And, by the column-sparsity of $M'$, any $x_i$ has degree at most 3 in $g$, and so $g$ can be interpretted as a categorical generating polynomial for $k = 4$ categories. By this interpretation, the coefficient of the monomial $\prod_{i=1}^{n'} x_i$ in $g$ is the probability of the assignment $X_1 = X_2 = \ldots = X_{n'} = 1$, a single likelihood, completing our reduction.

This motivates the need to research tractable categorical distributions, for example, possibly in the direction suggested by Cao et al. [2023]. In particular, this calls for careful consideration of the use of generating functions over categorical variables, which are not tractable models in general.

# 8 CONCLUSION

We studied tractable probabilistic circuits computing various polynomial representations of probability distributions. For binary probability distributions we show that a number of previously studied polynomials have equally expressive-efficient circuit representations. Among circuits computing

network, likelihood, generating, and Fourier polynomials, all support tractable marginal inference, and, given a circuit computing any one polynomial, a circuit computing any other can be obtained with at most a polynomial increase in size. This establishes a relationship between several previously-independent marginal inference algorithms, and establishes one novel marginal inference algorithm, namely for circuits computing likelihood polynomials. These results connect well-studied mathematical objects like generating functions and Fourier transforms in their forms as tractable probabilistic circuits, opening up potential future research, for example leveraging theory developed in one semantics and translating it to another, or learning in one representation space and transforming to another.

## Acknowledgements

We thank Benjie Wang, Poorva Garg, and William Cao for helpful comments on this work. This work was funded in part by the DARPA PTG Program under award HR00112220005, the DARPA ANSR program under award FA8750-23-2-0004, and NSF grants #IIS-1943641, #IIS-1956441, #CCF-1837129.

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

# A   PROOFS

Proof of Theorem 5.

*Proof.* Let $p(x)$ be a decomposable circuit of size $s$ computing a likelihood polynomial. We construct a circuit computing $\hat{p}(x)$ inductively as follows. For a product node we have $p(x) = q(x_q)r(x_r)$ where $x_q$ and $x_r$ partition $x$ and with $x_q$ of dimension $0 \le d \le n$. We have

$$\hat{p}(x) = 2^{-n} \sum_{v \in \{0,1\}^n} p(v)(-1)^{\langle v,x \rangle} \tag{7}$$

$$= 2^{-n} \sum_{v \in \{0,1\}^n} q(v_q)r(v_r)(-1)^{\langle v_q,x_q \rangle}(-1)^{\langle v_r,x_r \rangle} \tag{8}$$

$$= \left( 2^{-d} \sum_{v_q \in \{0,1\}^d} q(v_q)(-1)^{\langle v_q,x_q \rangle} \right) \cdot \left( 2^{d-n} \sum_{v_r \in \{0,1\}^{n-d}} r(v_r)(-1)^{\langle v_r,x_r \rangle} \right) \tag{9}$$

$$= \hat{q}(x_q)\hat{r}(x_r) \tag{10}$$

where the first equality follows from definition, the second from the hypothesis, the third by factoring, and the final from definition. For a sum node, we have $p(x) = \sum_i w_i p_i(x)$, and so

$$\hat{p}(x) = 2^{-n} \sum_{v \in \{0,1\}^n} p(v)(-1)^{\langle v,x \rangle} \tag{11}$$

$$= 2^{-n} \sum_{v \in \{0,1\}^n} \left( \sum_i w_i p_i(v) \right)(-1)^{\langle v,x \rangle} \tag{12}$$

$$= \sum_i w_i 2^{-n} \sum_{v \in \{0,1\}^n} p_i(v)(-1)^{\langle v,x \rangle} \tag{13}$$

$$= \sum_i w_i \hat{p}_i(x) \tag{14}$$

where the equalities follow, respectively, from definition, hypothesis, commutativity of addition, and definition.

For leaf nodes, it suffices to consider only univariate leaves that are children of sums; for any leaf a child of a product node, add a sum node with weight 1 between them. Then, for a univariate child of a sum node with scope the singleton $\{i\}$, we have either $p(x_i) = c$ for constant $c \in \mathbb{R}$, and so $\hat{p}(x_i) = 2^{-n}(c + c(1 - 2x_i))$ or $p(x_i) = x_i$, in which case $\hat{p}(x_i) = 2^{-n}(1 - 2x_i)$. □

Proof of Theorem 6.

*Proof.* We provide a deterministic, polynomial-time reduction from $\{0,1\}$-permanent to likelihood computation on a categorical PGC with $k = 4$ categories. Let $M \in \{0,1\}^{n \times n}$. Our reduction proceeds in two steps. We first 'sparsify' the matrix $M$ by finding a (modestly) larger matrix $M'$ with the property that every column of $M'$ contains at most three nonzero entries, while not changing the permanent: i.e. per $M$ = per $M'$. Then, we construct a simple circuit[5] using the entries of $M'$ which computes a polynomial $g$ such that the coefficient of a certain monomial in $g$ is exactly per $M$ = per $M'$. The column-sparsity of $M'$ guarantees that the degree of $g$ is at most 3, making it a valid generating function for $k = 4$ categories, and the desired coefficient is a particular likelihood.

**Step 1:** Suppose the $t$th column of $M$ contains more than three nonzero entries; if there is no such column, proceed to the second step. Intuitively, to 'sparsify' $M$ we can append a new $(n + 1)$th row and column to the matrix, setting their main diagonal entry to 1 as well as the $t$th entry of the new row – all other entries zero. Then, we are free to 'move' nonzero entries from the $t$th column to the new $(n + 1)$th column without affecting the permanent. For any nonzero term in per $M = \sum_\sigma \prod_i^n M_{i,\sigma(i)}$, there is a corresponding nonzero term in per $M' = \sum_\sigma \prod_i^{n+1} M'_{i,\sigma(i)}$, and no new nonzero terms have been introduced.

---

[5]This polynomial has appeared before in [Valiant, 1979a, Koiran and Perifel, 2007].

We now show this explicitly. Let $M \in \{0,1\}^n$ and assume the $t$th column of $M$ has more than three nonzero entries. Let $M_{a,t} = M_{b,t} = 1$ with $a \neq b$ be two of the nonzero entries in the $t$th column. Form a new matrix which is simply $M$ but with these two entries set to zero:

$$M_{i,j}^* = \begin{cases} 0 & i \in \{a,b\} \text{ and } j = t \\ M_{i,j} & \text{else} \end{cases} \tag{15}$$

We now define the new $(n+1)$th row and column $r, c \in \{0,1\}^n$. Set $r_t = 1$ and for $j \neq t$ set $r_j = 0$. Set $c_a = c_b = 1$ and for $j \notin \{a,b\}$ set $c_j = 0$. We now form the new matrix $M'$:

$$M' = \left( \begin{array}{ccc|c} & & & c_1 \\ & \mathbf{M}^* & & \vdots \\ & & & c_n \\ \hline r_1 & \cdots & r_n & 1 \end{array} \right) \tag{16}$$

To show that $\operatorname{per} M = \operatorname{per} M'$, we provide a bijection between the nonzero monomials of $\operatorname{per} M = \sum_\sigma \prod_i^n M_{i,\sigma(i)}$ and the those of $\operatorname{per} M' = \sum_\sigma \prod_i^{n+1} M'_{i,\sigma(i)}$ (viewing the monomials as formal objects). Recall that $\operatorname{per} M$ is the sum

$$\operatorname{per} M = \sum_{\sigma \in S_n} \prod_{i=1}^n M_{i,\sigma(i)} \tag{17}$$

Because $\sigma$ is a bijection, every term $\prod_{i=1}^n M_{i,\sigma(i)}$ in this sum contains $M_{i,t}$ for some $0 \leq i \leq n$. If $i \notin \{a,b\}$ we map

$$M_{1,\sigma(1)} \cdots M_{n,\sigma(n)} \mapsto M_{1,\sigma(1)} \cdots M_{n,\sigma(n)} M_{n+1,n+1}. \tag{18}$$

If $i \in \{a,b\}$ we map

$$M_{1,\sigma(1)} \cdots M_{i,\sigma(i)=t} \cdots M_{n,\sigma(n)} \mapsto M_{1,\sigma(1)} \cdots M_{n+1,t} \cdots M_{n,\sigma(n)} M_{i,n+1}. \tag{19}$$

This map is injective by construction. To show that it is also surjective, consider an arbitrary nonzero term $m = \prod_i^{n+1} M'_{i,\sigma(i)}$ in $\operatorname{per} M' = \sum_\sigma \prod_i^{n+1} M'_{i,\sigma(i)}$. Since $M_{n+1,t}$ and $M_{n+1,n+1}$ are the only two nonzero entries of the $(n+1)$th row of $M$, one of them must appear in $m$. If $M_{n+1,n+1}$ is in $m$, then the the monomial obtained by removing $M_{n+1,n+1}$ from $m$ is the preimage of $m$ by Eq. 18. If $M_{n+1,t}$ appears in $m$, then some other nonzero entry of the $(n+1)$th column must appear in $m$, namely $M_{i,n+1}$ for $i \in \{a,b\}$. So remove $M_{i,n+1}$ and $M_{n+1,t}$ from $m$ and inserts $M_{i,t}$ to obtain the preimage of $m$ by Eq. 19. So, we have $\operatorname{per} M = \operatorname{per} M'$.

Observe that the number of nonzero entries in column $t$ has decreased by one, and the number of nonzero entries in the new $(n+1)$th column is three. Repeat this step until all columns contain at most three nonzero entries, requiring at most $n^2$ repetitions.

**Step 2:** Call the matrix resulting from the first step $M' \in \{0,1\}^{n' \times n'}$ (where $n' \leq n + n^2$). Now consider the polynomial

$$\begin{aligned} g(x_1, \ldots, x_{n'}) &= \prod_{i=1}^{n'} \sum_{j=1}^{n'} M'_{i,j} x_j \\ &= (M'_{1,1} x_1 + M'_{1,2} x_2 + \ldots + M'_{1,n'} x_{n'}) \\ &\quad \cdot (M'_{2,1} x_1 + M'_{2,2} x_2 + \ldots + M'_{2,n'} x_{n'}) \\ &\quad \vdots \\ &\quad \cdot (M'_{n',1} x_1 + M'_{n',2} x_2 + \ldots + M'_{n',n'} x_{n'}) \end{aligned}$$

where $M'_{i,j} \in \{0,1\}$ are the entries of $M'$. Note that this expression for $g(x)$ directly provides a polynomial size circuit for $g(x)$ (in fact, a formula: a circuit whose DAG is a tree). Observe that the coefficient of the monomial $\prod_{i=1}^{n'} x_i$ in $g(x)$ is exactly $\operatorname{per} M' = \operatorname{per} M$. Moreover, the degree of any $x_i$ in $g(x)$ is at most three by the column-sparsity of $M'$, and so $g(x)$ can be interpreted as a categorical PGC with $k = 4$ categories. Given this interpretation, the coefficient of the monomial $\prod_{i=1}^{n'} x_i$ in $g(x)$ (which is $\operatorname{per} M$) is the probability of the assignment $X_1 = X_2 = \ldots = X_{n'} = 1$, a single likelihood.

$\square$