# OpenReview forum: "Polynomial Semantics of Tractable Probabilistic Circuits"
_auai.org/UAI/2024/Conference — UAI 2024 oral_

### Official Review · Reviewer_z2Dp · 2024-03-06

**Q2-1 Originality-Novelty:** 3
**Q2-2 Correctness-Technical Quality:** 4
**Q2-5 Clarity Of Writing:** 3

**Q1 Summary And Contributions:**

The paper focuses on the relationships of different types of tractable probabilistic models that represent a probability distribution of binary random variables as a circuit. The interest to such models stems from that probabilistic inference like marginal queries can be performed with them in a polynomial time in the size of the model and the query. The main contribution of the paper is showing that many seemingly different model families can be transformed into each other with only a polynomial increase in the size of the model, proving that the families are equally expressive efficient. Additionally, the paper considers a generalization of probability generating circuits for encoding categorical distributions and shows that even likelihood queries are #P-hard with them.

**Q2-3 Extent To Which Claims Are Supported By Evidence:**

3: Good: the main claims are supported by convincing evidence (in the form of adequate experimental evaluation, proofs, (pseudo-)code, references, assumptions).

**Q2-4 Reproducibility:**

4: Excellent: key resources (e.g. proofs, code, data) are available and key details (e.g. proof sketches, experimental setup) are comprehensively described for competent researchers to confidently and easily reproduce the main results.

**Q3 Main Strengths:**

The paper significantly advances the theory of tractable probabilistic models by showing that all the discussed tractable probabilistic models are roughly equivalent. Consequently, future results for one of the model families can be applied to all of them.

Additional theoretical contributions are provided in exploring how far the models can be extended until probabilistic inference becomes intractable with them, in this case by studying categorical distributions.

**Q4 Main Weakness:**

The Authors essentially prove that all discussed circuits can be reduced into a probability generating circuit (PGC) with only a polynomial increase in the size of the circuit, and vice versa. On the other hand, it is NP-hard to test if a given PGC encodes a valid (possibly unnormalized) probability distribution [Harviainen et al., 2023], i.e., all coefficents are nonnegative. To my understanding, this suggests that learning the structure and the weights for any of the discussed model families is an NP-hard problem, which I consider a weakness in terms of applying the results to practice. Please correct me if I have misunderstood.

**Q5 Detailed Comments To The Authors:**

Occasionally, the mathematical expressions could be made clearer by adding parentheses: e.g., after Proposition 2 there is an expression of the form $\prod (x + \bar{x}) \sum \prod \prod$, but to avoid confusion, $\left(\prod (x + \bar{x})\right) \sum \prod \prod$ might be clearer. Additionally, that specific line exceeds page margins.

Possible typos:
- I think Eq. (7) should have $p(x)$ instead of $p(x_S)$, since $x_S$ refers to an assignment of the random variables to my understanding.
- In Eq. (11), should the summands be $M_{ij}'x_j$ instead of $M_{ij} x_j$? (And if so, you could denote the size of $M'$ by $n'$). Of course, the specific coefficient matches the permanent even in the current form, but the degree of each $x_j$ can be $n$ instead of $3$ if $M$ is used instead of $M'$.
- Consequently, Theorem 6 should probably contain $k \ge 3$ instead of $k > 3$?

**Q9 Complying With Reviewing Instructions:**

Yes

---

> ### Author Rebuttal · Authors · 2024-04-04
>
> Thank you for your review.
>
> Indeed, it is NP-hard to check if a circuit encodes a valid PGC, which implies that it is hard in general to check the validity of the semantics of any of these models. While a negative result, this is an inherent property of this class of most expressive-efficient tractable models known. In particular, our work motivates the need for further research into learning algorithms that capture a largest-possible fragment of such circuits, i.e. without restricting to, say, nonnegative weights. While it is NP-hard in general to guarantee valid semantics, there are sufficient conditions for validity known in individual models, which now by our results, imply sufficient conditions to learning valid models in the others. For example, decomposability and monotonicity (nonnegative weights) of a circuit computing a network polynomial guarantees validity, and recent work [1] incorporates negative weights while maintaining validity, e.g. by squaring structured-decomposable circuits to guarantee nonnegative outputs. So, yes this is in some sense a negative result, but we have established the equivalence of these seemingly disparate languages and in doing so motivated the need for general learning methods and enabled the porting of results between the languages.
>
> [1] Lorenzo Loconte, Aleksanteri Mikulus Sladek, Stefan Mengel, Martin Trapp, Arno Solin, Nicolas Gillis, Antonio Vergari. Subtractive Mixture Models via Squaring: Representation and Learning. ICLR 2024.
>
> We now address your detailed comments below.
>
> [clearer expressions]
> - Very good point, thank you. We will make this change.
>
> [eq 7 notation]
> - You are correct that we did not define the notation for plain elements of $\{0,1\}^n$ (only for assignments to random variables), and we will do so here since we do mean the element of $\{0,1\}^n$ identified naturally with $S$.
>
> [eq 11 summands]
> - Yes, and yes. We will update; thank you.
>
> [$k\ge 3$ vs. $k>3$]
> - Interestingly, our current approach only guarantees that the number of entries of each column is at most $3$ which in turn produces a polynomial of degree at most $3$, but since the generating function also uses degree $0$, this only settles the case of $k>3$ (or $k\ge 4$).

---

### Official Review · Reviewer_EGPS · 2024-03-11

**Q2-1 Originality-Novelty:** 3
**Q2-2 Correctness-Technical Quality:** 3
**Q2-5 Clarity Of Writing:** 4

**Q1 Summary And Contributions:**

The paper creates a bridge among several semantics for Probabilistic Circuits based on a polynomial representation, proving that, for binary distributions, they are equivalent.

**Q2-3 Extent To Which Claims Are Supported By Evidence:**

2: Fair: the main claims are somewhat supported by evidence (but the experimental evaluation may be weak, or does not match entirely with the claims, important baselines may be missing, proofs contain important ideas but lack rigor, algorithmic details are only discussed superficially, references are imprecise, assumptions are not sufficiently motivated or explicated, etc.).

**Q2-4 Reproducibility:**

2: Fair: key resources (e.g. proofs, code, data) are unavailable but key details (e.g. proof sketches, experimental setup) are sufficiently well-described for an expert to confidently reproduce the main results.

**Q3 Main Strengths:**

The paper provides a link among several semantics, which is a very good point in establishing the equivalence of the expressive power. Furhtermore, it is well written.

**Q4 Main Weakness:**

In some parts the paper is not fully clear and requires to check many related works.
The usefulness of the presented work is not directly clear.

**Q5 Detailed Comments To The Authors:**

In my opinion, the paper is interesting but I think I’ve missed something. In particular, the paper should be more clear on the usefulness of its approach. While establishing connections among different representations is often important, here I do not see the advantages of a polynomial representation of PC and the advantages that a connection among polynomial representations yields. This is not a criticism on the usefulness per se of the paper (which is interesting) but on the advantages that converting a probabilistic circuit (PC) into a polynomial provides. I think this is a point that the paper should clearly state. For example, as far as I have understood, to obtain a polynomial from a PC, the circuit should be traversed. Once traversed, the obtained polynomial can be evaluated to compute the probability of a query. Now, what are the advantages of computing the probability of a query by extracting a polynomial from a PC and then evaluating it rather than, as usual, compute the probability by traversing it? With the standard approach, once we reach the root of the PC we have the probability of the considered query. On the other hand, if we extract a polynomial, we need to also evaluate it, so inference will be slower. Are there some reasoning steps that can be applied on the polynomial that are not evident from the PC? Again, this should be discussed and clearly written in both cases, i.e., whether there are advantages in terms of performance or it is ‘only’ a theoretical contribution (which is, again, completely fine).

I’ve other minor comments:
- the introduction should spend some more words on the structure of the paper, rather than only describing the content of section 6 and 7
the background should be extended with some examples of, for example, multilinear polynomials
- Last paragraph of section 2: is the condition that two circuits agree on all inputs on R^n stronger than agreeing on {0,1}^n, i.e., does the former include the latter? I think so, but I suggest specifying it
- end of page 3, above the last paragraph: is the flat representation less succinct than the circuit representation, right? I suggest specifying it
- below proposition 2: is this a proof? If so, I suggest enclosing it into a proper environment. The same consideration applies for the unlabeled equation after (3)
- above eq 4, in the text: what is H_0[f] ?

Overall, the paper is interesting but it should be integrated with the motivation of the proposed approach and some examples. There is more than one column of space left, so the new material should not require deleting something else.

**Q9 Complying With Reviewing Instructions:**

Yes

---

> ### Author Rebuttal · Authors · 2024-04-04
>
> Thank you for your review.
>
> Your questions highlight excellently an opportunity for us to improve the exposition of our work, in particular to clarify the relationship between the circuit and the polynomial; this matter lies at the core of our paper. You are correct that standard PC inference requires a simple forward pass of the circuit on a given input. However, if you hypothetically were to perform a forward pass on symbolic inputs (i.e. indeterminates $x_1,x_2,\ldots,x_n$), then a polynomial is computed (since the symbols are combined only via sums and products). In this sense, every circuit expresses some polynomial. Moreover, while the circuit may be small (say polysize in the number of variables), the polynomial it computes may be exponential in size (e.g. the network polynomial for the uniform distribution has exponential size when written as a “flat” polynomial while it can be expressed very concisely as a circuit). So, a circuit allows us to express an exponential size polynomial succinctly.
>
> The idea of our paper is to study the relationship between circuits which express different polynomials. In particular, we show that a circuit expressing a generating polynomial can be transformed efficiently into a circuit expressing a network polynomial even though the polynomials need not ever be explicitly computed. This means that all inference algorithms considered in this work are efficient in the size of the circuit (not in the size of the expressed polynomial). In particular, our results hold not as “only theoretical” properties of the various polynomials, but as algorithms that operate on circuits, i.e. standard PCs.
>
> We hope this helps address your concerns and that you are able to update your review accordingly. If you have further questions, we are happy to discuss them.
>
> Concretely, we intend to add discussion along these lines to clarify the relationship between a circuit and the polynomial which it expresses, towards the end of Section 2.
>
> We also address your additional minor comments below.
>
> [more structure and examples in intro/background]
> - We agree and will modify the introduction and background accordingly, as you suggest including examples of multilinear polynomials.
>
> [Last paragraph of section 2, stronger condition]
> - Yes it is stronger, e.g. because $x=x^2$ on $\{0,1\}$ but not on $\mathbb{R}$. We will mention this.
>
> [end of page 3, above the last paragraph: flat is less succinct]
> - You are correct. We will clarify this in the wording.
>
> [below proposition 2: is this a proof?]
> - Yes, we will wrap it accordingly; thank you.
>
> [above eq 4, in the text: what is H_0[f] ?]
> - We will clarify the definition; $H_i[f]$ is the sum of all monomials in $f$ with degree exactly $i$. So, $H_0[f]$ is the constant term.

---

### Official Review · Reviewer_Tzs3 · 2024-03-20

**Q2-1 Originality-Novelty:** 2
**Q2-2 Correctness-Technical Quality:** 3
**Q2-5 Clarity Of Writing:** 3

**Q1 Summary And Contributions:**

This paper  mainly presents a series of polynomial transformations between different representations of a probability distribution by an arithmetic circuits;  the common point of all the   types of arithmetic circuit represented satisfy in polytime the marginalisation transformation

**Q2-3 Extent To Which Claims Are Supported By Evidence:**

2: Fair: the main claims are somewhat supported by evidence (but the experimental evaluation may be weak, or does not match entirely with the claims, important baselines may be missing, proofs contain important ideas but lack rigor, algorithmic details are only discussed superficially, references are imprecise, assumptions are not sufficiently motivated or explicated, etc.).

**Q2-4 Reproducibility:**

3: Good: key resources (e.g. proofs, code, data) are available and key details (e.g. proofs, experimental setup) are sufficiently well-described for competent researchers to confidently reproduce the main results.

**Q3 Main Strengths:**

The transformations presented are interesting and can be considered as a part   of a comprehensive KC map of arithmetic circuit.  The proof sketch of the original results are included in the paper.

**Q4 Main Weakness:**

The framework is limited to distributions over boolean variables

Experiments are missing that could compare the practical  (and relative) efficiency of the different types of circuits considered (and to more compact ones)

The reader misses also some results of succinctness with respect to decomposable forms and  to sentential diagrams

**Q5 Detailed Comments To The Authors:**

A succinctness map could be included in the paper, comparing the AC considered in the paper to other types of arithmetic circuits, and  especially to more compact ones. The authors should have   in particular a llok to:


Alexis de Colnet, Stefan Mengel:
A Compilation of Succinctness Results for Arithmetic Circuits. KR 2021: 205-215
(by the way it seems to me that this paper should be cited)

For instance, are decomposable circuits *strictly* more succinct  than the ones considered in the paper ? same question with respect to sentential diagrams

Can  the results  presented here be extented to non boolean variables ? how ?

**Q9 Complying With Reviewing Instructions:**

Yes

---

> ### Author Rebuttal · Authors · 2024-04-04
>
> Thank you for your review.
>
> [succinctness comparisons]
>
> First, we would like to mention that it has been shown in [1] that circuits representing probability generating polynomials (PGCs) are not only tractable (for marginals) but also strictly more succinct than the monotone decomposable circuits including PSDDs. One of the main contributions of our work is showing that circuits representing probability mass functions (and their Fourier transforms, and network polynomials) are equally succinct as PGCs, thus implying that the circuits considered in our work are strictly more succinct than the monotone decomposable circuits. We agree that a detailed discussion on succinctness is important and we will make it more clear in our paper.
>
> Thank you for referring to [2]. We note that while [2] studies the succinctness of circuits subject to various syntactic constraints, our work makes no assumptions on syntactic properties of the circuits (i.e. our transformations hold for arbitrary arithmetic circuits) and instead focuses on the semantics of the polynomials that they represent. One interesting connection is that, from the perspective of relaxing structural constraints, [2] considers consistency as a relaxation of decomposability and shows that a monotone and consistent yet not smooth circuit is strictly more succinct than monotone decomposable circuits. However, it is unclear whether inference with a consistent but not smooth circuit is tractable. We will include a more detailed discussion on the connection between [2] and our work.
>
> [limit to boolean variables and the extension beyond]
>
> While we would like to characterize the tractability of arbitrary distributions, we note that solving the boolean case is a prerequisite for doing so. Moreover, in Section 7 we provide the first negative result towards the question of tractable modeling of categorical variables, namely that inference in circuits representing generating functions for categorical variables is #P-hard. We also note that we can encode categorical random variables using binary indicator variables, as is standard in the literature. Finally, we recall that recent progress in the area of Probabilistic Circuits including modeling heterogeneous (discrete and continuous) data [3] was originally enabled by insights in the boolean setting, tracing back to early work on DNNF.
>
> [practical implications]
>
> A main contribution of our work is to show that circuits representing network polynomials, probability generating functions and their Fourier transforms are equally succinct, that is you can transform one into the other with at most polynomial increase in circuit size. Even though our algorithms are not meant to transform circuits to ones of smaller sizes, they do have significant practical implications. For example, even though it has been shown PGCs are strictly more expressively efficient, they cannot be easily implemented using the existing libraries for probabilistic circuits (e.g. EinsumNetworks and Juice.jl), which assume the circuits represent network polynomials. Our algorithm that transforms PGCs to circuits representing network polynomials immediately opens a new avenue for deploying and scaling up PGCs in practice. Moreover, we establish a novel inference algorithm for circuits computing likelihood polynomials that need not be smooth, saving up to a quadratic blowup in standard decomposable PCs in inference time. As Reviewer zZNh writes, “Theorem 2 as a generalization for smoothing could see broad applicability.” We agree that eventually it is important to explore the use of circuits studied in our work as practical machine learning models, which is also ongoing research, but before that we need to establish the theoretical foundations, which is the main focus of this work.
>
> [1] Honghua Zhang, Brendan Juba, and Guy Van den Broeck. Probabilistic generating circuits. In Proceedings of the 38th International Conference on Machine Learning (ICML), 2021.
>
> [2] Alexis de Colnet, Stefan Mengel: A Compilation of Succinctness Results for Arithmetic Circuits. KR 2021: 205-215.
>
> [3] Z. Yu, M. Trapp, K. Kersting. Characteristic Circuits, 37th Conference on Neural Information Processing Systems, 2023.

---

### Official Review · Reviewer_43Yz · 2024-03-22

**Q2-1 Originality-Novelty:** 3
**Q2-2 Correctness-Technical Quality:** 4
**Q2-5 Clarity Of Writing:** 3

**Q1 Summary And Contributions:**

This paper studies probabilistic circuits, and a variant where also division nodes are present, as a means to efficiently compute useful functions based on probability distributions. The authors' starting point is a network polynomial, which is a certain polynomial transform of a joint probability function. They show how likelihood polynomials, generating polynomials and Fourier transforms can be efficiently calculated by transforming a probabilistic circuit that calculates network polynomials into one that calculates the other types of functions. Their main assumption so far is that each random variable is binary.

In the second part of the paper, the authors go on and show that no efficient calculation is possible when the random variables are no longer binary.

**Q2-3 Extent To Which Claims Are Supported By Evidence:**

4: Excellent: all claims are supported by very convincing evidence (in the form of comprehensive experimental evaluation, rigorous mathematical proofs, detailed (pseudo-)code, precise references, well-motivated and realistic assumptions) and the authors deliver what they promise.

**Q2-4 Reproducibility:**

4: Excellent: key resources (e.g. proofs, code, data) are available and key details (e.g. proof sketches, experimental setup) are comprehensively described for competent researchers to confidently and easily reproduce the main results.

**Q3 Main Strengths:**

This paper is very well written. It introduces the concepts in a clear way, use natural notation (although their is some slight sloppiness – more about this in my Detailed Comments To The Authors), and has a logical structure. All this results in an enjoyable and educational read.

The content is interesting and relevant. The first sections 3 – 6 are important and well explained. I like the idea at the end of Section 3.2 to decompose a circuit with division nodes to a standard circuit. The authors use important results from earlier work, interpret them to the current context, and cite them well.

The final section 7 seems equally important to the first sections, if not more important. I do think that this is significantly less clearly explained (perhaps due to the page limitations); more about this in the Main Weaknesses part.

As the paper deals with tractable ways of calculating polynomial and Fourier transforms of probability mass functions, the subject falls squarely into the subject of this conference.

Overall I think this is a good paper which will likely lead to interesting discussion.

**Q4 Main Weakness:**

I only identify two weaknesses.

In Section 7 the authors purport to show that computing likelihoods on categorical circuits with 4 or more (but finite) categories is #P-hard, by constructing a circuit whose likelihood is calculated in the same way as the permanent of a matrix.
My concern with this section is in Step 1. The authors do not sufficiently explain how to calculate M' from M. The figure 3 is helpful, but significantly more details on this procedure are necessary. For instance, what happens if I start with the zero matrix 0^{n x n}? Then the sentence "Now select any two of the original nonzero entries ..." is void. Should this sub-step then be skipped? If so, who do we end up with a matrix of which all its columns contain at most three nonzero entries?

Even if this cannot be made sufficiently clear, I don't think this would hamper the paper too much, as Section 7 can simply be omitted (as the paper has enough content in the prior sections, to my estimation), or its content stated as a conjecture, and the arguments for it described as a sketch.

As a second, minor, weakness I identify the lack of experimental results. I do believe, however, that this is not at all necessary for this paper, as (i) it explains theoretical concepts, (ii) the paper provides proofs for results which no doubt will lead to applications, and (iii) the paper has more than enough content already, but I realise that some people might have a different opinion.

**Q5 Detailed Comments To The Authors:**

Here are my detailed comments.



Page 2, Definition 1, Part 1: I would mention here, or in Figure 2, that weights of 1 are not displayed.

Page 2, final sentence of Section 2, "... the polynomials are equivalent elements ...": What are equivalent polynomials? Unless I misunderstand the content, I think that "identical" would be better here.

Page 3, Figure 2 parts b and c: $\overline{X_1}$ and $\overline{X_2}$ should be $\overline{X}_1$ and $\overline{X}_2$, to be consistent with the notation later on, for instance in Equation (1).

Page 3, Proposition 1: I would mention that we are marginalizing OUT $X_i$, instead of marginalizing TO, which is how I initially read it. I realize that the current terminology is standard in some literature, but I was not immediately aware of it.

Page 4, second equation of the left column: $\mathbf{x}_s$ should read $\mathbf{x}_S$, which is how it is defined in Section 2. In fact, this is not consistently used throughout the paper: Equation (7), the equation at the bottom of Page 6, and the following text use $x_S$; the proof of Proposition 3 uses $\mathbf{X}_s$ and $\mathbf{X}_S$.

Page 4, line above Equation (4): The index $j$ should start at $0$ instead of $1$.

Page 4, paragraph following Equation (4): Why use the authors $\overline{p}$ to indicate the network polynomial here, instead of $p$ as they defined it? On Page 5 they use also $\overline{p}$. I suggest they make this uniform.

Page 7, Proof of Theorem 6: The authors say that they prove Theorem 6 by reducing the computation of a permanent, which is known to by #P-hard, to the computation of the likelihood. However, this would not constitute a proof of Theorem 6, I believe: this would only show that the computation of the likelihood cannot be harder than #P-hard, but it leaves open that it is potentially easier. I do believe, however, that this is not what the authors show (provided the transformation of M into M' is correct, which lacks details to check this): they map any calculation of a permanent to a likelihood calculation, which is indeed as wanted, so (again provided that Step 1 is correct) this constitutes indeed a correct proof. I suggest that the authors change this first sentence.

Page 8, sentence following Equation (11): There is a redundant 'the'.

**Q9 Complying With Reviewing Instructions:**

Yes

---

> ### Author Rebuttal · Authors · 2024-04-04
>
> Thank you for your review.
>
> We agree that the first step in the proof of Theorem 6 can be made more clear. As you suggest, we will address the $0^{n\times n}$ case (here the first step should be skipped as each column already has $0\le 3$ nonzero entries).
>
> While we agree that there are exciting directions for experimental work opened by this paper, we ultimately decided that to do the theoretical results justice given the page limit, this paper should only contain the theory. As Reviewer zZNh put it, “the presentation of the material is [already] very streamlined considering the number of results packed into such a short paper.” Moreover, we agree with your three points.
>
> Response to Detailed Comments:
>
> We thank you for the several typos pointed out and give responses to comments calling for them below.
>
> Page 2, final sentence of Section 2, "... the polynomials are equivalent elements ...": instead use “identical”
> - Good point. We will change to “identical”.
>
> Page 3, Figure 2 parts b and c consistent notation
> - Yes, thank you.
>
> Page 3, Proposition 1: I would mention that we are marginalizing OUT
> - Very good suggestion, yes.
>
> Page 7, Proof of Theorem 6: direction of the reduction
> - While phrasing the ‘direction’ of a reduction is a common error, we have checked again and believe we have worded it correctly here.
>
> As for your comments Section 7, we are confident that we can (using the extra space) add further details to the first step of the reduction in Section 7 to make it more clear. This result is fundamental to the area of tractable probabilistic models since, before this paper, Probabilistic Generating Circuits were the most expressive-efficient model available for tractable marginal inference, and this result establishes their limits to binary variables.

---

### Official Review · Reviewer_zZNh · 2024-03-22

**Q2-1 Originality-Novelty:** 3
**Q2-2 Correctness-Technical Quality:** 4
**Q2-5 Clarity Of Writing:** 3

**Q10 Ethical Concerns:**

I don't foresee this work giving rise to any new ethical concerns.

**Q1 Summary And Contributions:**

This paper proposes a way to unify several methods of expressing probability distributions via probabilistic circuits (which in turn represent multilinear polynomials). In particular, they propose transformations between all the polynomial semantics in question (with nominal blowup in the size, which is alleviated in the presence of decomposability). This allows them to prove the tractability of computing marginals for all of them. On the intractability end of the spectrum, they also prove that, in the categorical setting (i.e., non-binary), extending the polynomial semantics is infeasible in most cases or in the case of PGCs, yields an intractable model.

**Q2-3 Extent To Which Claims Are Supported By Evidence:**

4: Excellent: all claims are supported by very convincing evidence (in the form of comprehensive experimental evaluation, rigorous mathematical proofs, detailed (pseudo-)code, precise references, well-motivated and realistic assumptions) and the authors deliver what they promise.

**Q2-4 Reproducibility:**

3: Good: key resources (e.g. proofs, code, data) are available and key details (e.g. proofs, experimental setup) are sufficiently well-described for competent researchers to confidently reproduce the main results.

**Q3 Main Strengths:**

The paper unifies several hitherto studied representations in an elegant manner. The timing of this paper also seems very apt due to the recent rise in popularity of this field of Probabilistic Circuits. I can imagine this paper having a big impact and helping to focus and maximize the output of any future effort. The presentation of the material is very streamlined considering the number of results packed into such a short paper.

**Q4 Main Weakness:**

The paper could be helped by a more solid "Background/Preliminaries" section. At times, I found myself having to assume and guess things in order to proceed, only to realize something and then be forced to backtrack. Admittedly, none of the techniques used in the paper are ground-breaking but in my opinion that doesn't diminish the value of the result.

**Q5 Detailed Comments To The Authors:**

Many of my comments refer to either missing or incomplete definitions/notation/background, which should be fairly straightforward to accomdate consdering that over a column of space is still left.

(mostly minor remarks and typos)

- [p3, Proposition 1] '.' missing at the end of Proposition 1
- [p3, right column, 2nd para] "transformation 2 in Figure 1", I think this should be transformation **3**
- [p4, Theorem 2] mention the corresponding transformation (transformation 4)
- [p4, right column, 4th para] $\sum_{j=1}^\infty$ should be $\sum_{j=0}^\infty$
- [p4, right column, 5th para] what's $\bar p$? I couldn't find it's definition
- [p4, right column] the ending of Section 3 feels a bit rushed, and could be made more gentle. a little more background on "homogenization" would've been nice, e.g. define "homogenous", $\mathrm H_i[f]$ has degree $i$ **and is homogenous**
- [p5, left column, after Eqn (5)] "Zhang et al. [2021] ... show marginal inference is tractable; this is transformation 2" it is unclear what this refers to in the cited paper. It seems to imply that transformation 2 was used to show tractability but that doesn't seem correct.
- [p5, Theorem 3] mention the corresponding transformation (transformation 1)
- [p5, right column, 2nd para] "supporting marginal inference when the circuit **is** smooth"
- [p5, Eqn (6)] inside the summation should it be $p(\boldsymbol x)$ instead of $p(x)$?
- [p5, Eqn (6)] the notation $\langle \boldsymbol t, \boldsymbol x \rangle$ is not defined
- [p5, after Eqn (6)] it is not obvious how to get from Eqn (6) to Eqn (7)
- [p5, Eqn (7)] inside the summation should it be $p(\boldsymbol x_S)$ instead of $p(x_S)$?
- [p6, proof of Proposition 3] should it be lowercase $\boldsymbol x_S$ instead of $\boldsymbol X_S$ inside $\text{Pr}$?
- [p6, Theorem 4] "generating polynomial $g$ **(resp. $\hat p$)** for Pr ~(resp. $\hat p$)~ ... Fourier transform $\hat p$ **(resp. $g$)** for Pr ~(resp. $g$)~"
- [p6, left column, last para] bare references to $p$ are ambiguous and it's not clear if they refer to $p(x, \bar{x})$ or $p(x)$. This also holds for $\hat{p}$, $p_{-1, 1}$.
- [p6, left column, last para] "it is more common to define boolean Fourier transform ...", is there a specific reference for this? maybe Thm 1.1 from O'Donnel [2014]?
- [p6, left column, last para] should it be $\hat p(\boldsymbol x_S)$ instead of $\hat p(x_S)$?
- [p6, right column, 1st para] "transformations 5 and 7" should be "transformations 5 and 6" if I'm not mistaken
- [p6, right column, 1st para] unfortunately I couldn't follow the last 3-4 lines of this paragraph. How does the transformation from $g$ to $p(x, \bar x)$ help?
- [p6, right column, 3rd para] "scope of the node" should be in `\emph`
- [p6, Definition 3] "equal scope", I would use "identical" or "same" instead of "equal"
- [p6, right column, last para] "Theorems 2,3,4" spacing needs to be fixed
- [p7, sketch of Theorem 5] could expand a bit by showing one such 'push' of the Fourier transform
- [p7, Lemma 2] mention the reference of [Shih et al. 2019, Choi et al. 2020] next to the lemma number
- [p7, right column, 2nd to last para] "equivallently" (typo)
- [p8, left column, 1st para] couldn't/didn't verify the claim $\text{per} M = \text{per} M'$ but otherwise the reduction is fine

- (not a question, just a remark) Theorem 2 as a generalization for smoothing could see broad applicability
- (unrelated to content of the paper) I experienced a peculiar font-related glitch when I printed out the paper, where all the $\sum$ and $\prod$ symbols disappeared. In case the paper gets accepted, it might be good to beware of this bug during the camera submission.

**Q9 Complying With Reviewing Instructions:**

Yes

---

> ### Author Rebuttal · Authors · 2024-04-04
>
> Thank you for your thorough review.
>
> Any comment for which we do not directly reply below is one for which we agree with your point and will make the associated change. The remaining comments that elicit a more substantive response, we provide it below.
>
> [p4, right column, 5th para] what’s $\bar{p}$?
> - Legacy notation for $p(x,\bar{x})$; we will update.
>
> [p4, right column] ending of Section 3
> - Yes, we agree, and we will update accordingly.
>
> [p5, left column, after Eqn (5)]
> - This is a typo. The note “; this is transformation 2 in Figure 1” belongs two sentences later, directly following “and so any distribution with a polysize circuit computing its network polynomial also has a polynomial-size PGC.”
>
> [p5, Eqn (6)] notation, bold vs not bold
> - Yes you are correct; we will generally update this, i.e. not use bold at all, but just use plain lower case letters for elements of $\{0,1\}^n$ and for vectors of indeterminates taking values in $\{0,1\}^n$.
>
> [p5, Eqn (6) to (7)]
> - Good point, thank you. If we identify $S$ with $x_S$ (to switch what we’re summing over), then $(-1)^{\langle x_S,t\rangle}=(-1)^{\sum_{i\in S} t_i}=\prod_{i\in S}(-1)^{t_i}=\prod_{i\in S}(1-2t_i)$. We will add this to the paper.
>
> [p6, left column, last para] bare references to $p$
> - Yes. We will update all such instances.
>
> [p6, left column, last para] specific reference … maybe Thm 1.1 from O'Donnell?
> - Great suggestion. We will add this. (And yes, Thm 1.1 from O’Donnell is perfect.)
>
> [p6, right column, 1st para] couldn't follow the last 3-4 lines
> - We agree that the exposition of this point is currently far too terse; thank you for noting this. We have given an expanded version below (picking up after the equation that introduces the Fourier expansion), and we hope this clarifies the point.
>
> Here the values $\hat{p}(\cdot)$ are called the *Fourier coefficients*, or, collectively, the *Fourier spectrum* of $p(x)$. Moreover, this is more naturally written on the domain $\{-1,1\}^n$ due to the equivalence of the parity functions $\prod_{i\in S}(1-2x_i)$ to the monomials $\prod_{i\in S}x_i$ on the respective domains $\{0,1\}^n$ and $\{-1,1\}^n$.
> That is, we have
> $$p_{-1,1}(x) =\sum_{S\subseteq [n]}\hat{p}(v_S)\prod_{i\in S}x_i.$$
> (Here $v_S$ is the element of $\{0,1\}^n$ with $v_i=1$ for $i\in S$ and $v_i=0$ for $i\notin S$.)
> Now we recall the transformation from $g(x)$ to $p(x,\bar{x})$ (transformation 1 in Figure 1). In general, this transformation takes a polynomial in indeterminates $ x_1,\ldots,x_n $ with monomials $c_S\prod_{i\in S}x_i$ for $S\subseteq [n]$ and produces a polynomial in indeterminates $x_1,\ldots , x_n,\bar{x_1},\ldots ,\bar{x_n}$ with monomials $c_S\prod_{i\in S}x_i\prod_{i\notin S}\bar{x_i}$ for $S\subseteq [n]$; concretely, this latter polynomial allows us to extract the coefficients of the former by the substitution $\bar{x_i}=1-x_i$. Now looking at the previous equation, we have an identical problem where we would like to compute $\hat{p}(x)$ from $p_{-1,1}(x)$, i.e. to `extract' the Fourier coefficients from $p_{-1,1}(x)$. So, we analagously define the polynomial $\hat{p}(x,\bar{x})$ as
> $$\hat{p}(x,\bar{x}) =\sum_{S\subseteq [n]}\hat{p}(v_S)\prod_{i\in S}x_i\prod_{i\notin S}\bar{x_i}$$
> and obtain transformations 7 and 8 in Figure 1. Moreover, the relationship between $\hat{p}(x,\bar{x})$ and $\hat{p}(x)$ is identical to that between $p(x,\bar{x})$ and $p(x)$, yielding transformations 9 and 10 in Figure 1 (by Theorem 2).
>
>
> [p8, left column, 1st para] couldn't/didn't verify the claim $\text{per}M=\text{per}M’$ but otherwise the reduction is fine
> - We will elaborate. Any nonzero term in $\text{per} M$ can be seen as in direct (bijective) correspondence with a nonzero term in $\text{per} M$. The key is that the ‘original’ and ‘new’ columns of $M’$ both have a $1$ in the $i$th row; that is, $M_{i,i}'=M_{i,i+1}'=1$. Any nonzero term in $\text{per} M$ corresponds to a single $1$ in the $i$th column. In $M’$, that $1$ is either still in the $i$th column (in which case M_{i,i+1}' covers the $i$th row and gives us a nonzero term in $\text{per} M’$) or the $1$ was ‘moved’ to the $(i+1)$th column (in which case M_{i,i+1}' covers the $i$th row and gives us a nonzero term in $\text{per} M’$).
>
> (not a question, just a remark) Theorem 2 as a generalization for smoothing could see broad applicability
> - This is a great point, and we agree. In particular, while there has been effort towards reducing the cost of smoothing (e.g. by Shih et al. 2019 for structured circuits) since smoothness has to this point been considered a prerequisite for efficient marginal inference, now Prop 2 guarantees efficient (linear in circuit size) inference without smoothness. Moreover, Theorem 2 allows any circuit computing a multilinear mass function (without decomposability) to be made ‘smooth’, i.e. to compute a network polynomial.
>
> (unrelated to content of the paper) font-related glitch
> - Peculiar indeed; we’ll look into this.

---

### Meta-Review · Area_Chair_DGKR · 2024-04-16

Various tractable (circuit based) representations of binary probability distributions have been presented in the literature. The present paper studies the relationships of different representations, showing that many of them are equivalent in the sense that they can be transformed into each other with just a polynomial increase in the size. The paper also shows a negative result: a natural representation of non-binary categorical distributions renders even likelihood queries #P hard.

The paper was very well received by all the reviewers. The reviewers generally found the paper well written but also gave suggestions to improve the clarity of writing in some places.

The paper is likely to be of interest to a wider audience.